# MTR-SAM: Visual Multimodal Text Recognition and Sentiment Analysis in Public Opinion Analysis on the Internet

Xing Liu [1,2], Fupeng Wei [3,*], Wei Jiang [3], Qiusheng Zheng [1,2], Yaqiong Qiao [3,4], Jizong Liu [1,2], Liyue Niu [1,2], Ziwei Chen [1,2] and Hangcheng Dong [5,*]

1　The Frontier Information Technology Research Institute, Zhongyuan University of Technology, Zhengzhou 450007, China; xl@zut.edu.cn (X.L.); zqszut@163.com (Q.Z.); 31.jz@163.com (J.L.); 9715@zut.edu.cn (L.N.); czw@zut.edu.cn (Z.C.)
2　Henan Key Laboratory on Public Opinion Intelligent Analysis, Zhengzhou 450007, China
3　School of Information Engineering, North China University of Water Resources and Electric Power, Zhengzhou 450046, China; jiangwei@ncwu.edu.cn (W.J.); kitesmile@126.com (Y.Q.)
4　Henan Key Laboratory of Cyberspace Situation Awareness, Zhengzhou 450001, China
5　School of Instrumentation Science and Engineering, Harbin Institute of Technology, Harbin 150001, China
*　Correspondence: weifupeng@yeah.net (F.W.); hunsen_d@hit.edu.cn (H.D.)

**Abstract:** Existing methods for monitoring internet public opinion rely primarily on regular crawling of textual information on web pages but cannot quickly and accurately acquire and identify textual information in images and videos and discriminate sentiment. The problems make this a challenging research point for multimodal information detection in an internet public opinion scenario. In this paper, we look at how to dynamically monitor the internet opinion information (mostly images and videos) that different websites post. Based on the most recent advancements in text recognition, this paper proposes a new method of visual multimodal text recognition and sentiment analysis (MTR-SAM) for internet public opinion analysis scenarios. In the detection module, a LK-PAN network with large sensory fields is proposed to enhance the CML distillation strategy, and an RSE-FPN with a residual attention mechanism is used to improve feature map representation. Second, it proposes that the original CTC decoder be replaced with a GTC method to solve earlier problems with text detection at arbitrary rotation angles. Additionally, the performance of scene text detection for arbitrary rotation angles is improved using a sinusoidal loss function for rotation recognition. Finally, the improved sentiment analysis model is used to predict the sentiment polarity of the text recognition results. The experimental results show that the new method proposed in this paper improves recognition speed by 31.77%, recognition accuracy by 10.78% on the video dataset, and the F1 score of the multimodal sentiment analysis model by 4.42% on the self-built internet public opinion dataset (lab dataset). The method proposed provides significant technical support for internet public opinion analysis in multimodal domains.

**Keywords:** public opinion analysis; sentiment analysis; multimodal; text recognition

## 1. Introduction

　　People use online platforms such as Weibo, Zhihu, TikTok, and Kuaishou to communicate with social hotspots and express their opinions. In addition to traditional text narratives, images and short videos are becoming increasingly significant vehicles for conveying personal emotions due to their vivid and rich content and other attributes. On the other hand, internet robots and water armies create and circulate deceptive material by utilizing characteristics such as images and videos that are not easily detected as containing sensitive data. This allows them to circumvent traditional security measures. In the event that they are not discovered in a timely manner, they will inflict enormous social losses and create the risk of adverse public opinion.



First, we look at how single-modal video or image text recognition and sentiment analysis were applied in the past. In the phase of dataset preparation, the majority of earlier research assumed that images and videos were clipped. These methods, on the other hand, can only be used for a single sense of application situation, are restricted to short video fragments, and are not reliable for identifying unclipped videos or images. In the model pre-training stage, the conventional method of image-text recognition is a complex and time-consuming operation. In recent years, with the continuous diffusion of deep learning technology, RNN (recurrent neural network) [1] and image text recognition techniques such as attention-based mechanisms and variant models have been developed [2]. However, there are still problems with the wrong position of recognition, a low recall rate, attention bias, and choosing the right features. Recognizing text in the massive amount of video appearing on the web is a problem that encompasses multiple theories, techniques, and modules. The current research strategy for text recognition in most videos mainly contains modules on text detection, text extraction, text recognition, etc. [3]. The text detection module finds the text in each video frame and recognizes the region. The text recognition module takes the text from the region image and turns it into a binary text image that the recognition module can use to find the scene text in the video. With the development of neural network technology, machine learning and deep learning-based methods have emerged and attained excellent recognition results. Internet-based public opinion monitoring services are closely related to sentiment analysis research, and sentiment analysis research increasingly tends to differentiate sentiment in complex scenarios [4]. The emergence of deep learning technology increases the likelihood of accurate discrimination in sentiment analysis, comprehensively capturing text feature information, and achieving good experimental results. In contrast to single-modal studies, multimodal research is concerned with extracting global and local feature information from text, images, video, audio, etc. [5]. Currently, multimodal sentiment analysis is an important study direction, and the most typical application scenarios are network opinion situational awareness and intelligent companionship. The key to multimodal study is the integration of information across modalities. Previous studies generally fused information between modalities regardless of the importance of information between modalities, and the current fusion ignored the problem of sentiment inconsistency across modalities. Examples include CSID [6], CoMIR [7], AV-Robustness [8], etc.

Our lab has set up a project on internet opinion analysis to keep an eye on different websites in Henan Province, China for dangerous information in real time and to be able to send out timely warnings to protect the safe production of the network. Since information about terrorism and gambling is often put on websites, these postings can be videos, images, text, audio, etc. The goal of this paper is to monitor and discover images and videos linked to terrorism and gambling that are posted on these websites and to discover the public's opinion in real time. Technically, traditional internet-based techniques for monitoring public opinion rely on a single data source and the extraction of text data from web pages. In comparison to multimodal data, the current studies on algorithms for text recognition in videos and images lag behind. Existing methods for monitoring internet public opinion rely primarily on the regular crawling of text information on web pages, which is incapable of rapidly obtaining and recognizing text information in images and videos. In addition, current emotional analysis models cannot deal with the fact that text sequences can have more than one meaning and that information can be lost when intermediate vectors are used to encode features [9]. This can make it hard to understand the context of the modeled features. Based on the preceding, this paper combines the most recent advancements in text recognition with the characteristics of data in public opinion analysis. First, the PP-OCR [10] model deployed in the OpenVINO [11] environment is modified and adapted, and the improved method is able to solve the problem of text recognition in complex scenes, such as videos, while successfully resolving the shortcomings of the original model in terms of its inability to detect distorted images or videos. Then, a new multimodal text recognition and sentiment analysis model is proposed for detecting positive and negative information

from different dynamic websites with risky videos or risky images. The existing sentiment analysis model is modified, and the modified text recognition model is combined with the sentiment analysis model.

We propose the multimodal text recognition and sentiment analysis model MTR-SAM, which not only improves the most recent PP-OCR model but also introduces a direction-aware function that optimizes the recognition of rotated fonts and improves the detection of distortion and directional skew. This paper proposes a sentiment analysis model that improves the contextual relationship between words and the accuracy of discriminating the sentiment polarity of text. The proposed model for sentiment analysis can strengthen the contextual relationship between words and enhance the precision of text sentiment polarity discrimination. The experimental results show that the new method proposed in this paper improves the recognition speed by 31.77%, the recognition accuracy by 10.78% on the video dataset, and the F1 value of the sentiment analysis model by 4.42% on the user-generated opinion dataset. In addition, the language of the training set for the multimodal character recognition model is English and Mandarin, whereas the language of the training set for the sentiment analysis model is Mandarin. The model is primarily utilized to monitor Chinese websites, and the method can offer significant technical support for public opinion analysis in multimodal disciplines. Our contribution is in three areas:

- To address the problem of insufficient text detection models in existing multimodal scenes for large font text or text with an extreme aspect ratio. We proposed the LK-PAN network of large receptive fields in the detection module to improve the CML distillation strategy and then employed an RSE-FPN with a residual attention mechanism.
- As a solution to the issue of low accuracy of text recognition models for rotating fonts in existing multimodal scenes, we modified the original CTC decoder to use the GTC (Guided Training of CTC) method and applied a sinusoidal loss function for rotation recognition. This loss function can optimize the model's recognition of rotating fonts and improve text detection performance in settings with arbitrary rotation angles.
- In internet public opinion services, public opinion risk detection needs to be conducted on multimodal data, including images and videos. To distinguish the results of multimodal text recognition, we therefore introduce a sentiment analysis model. This model is capable of positive and negative sentiment discrimination on multimodal data such as images and videos. Additionally, this paper proposes a fusion of global and local information block attention mechanisms to solve the issue that word vectors are easily lost during the splicing process and the information loss of intermediate vectors during encoding.

## 2. Related Work

### 2.1. Public Opinion Scene Text Recognition

Text recognition is one of computer vision's most active areas of study [12]. Early Europe and the United States utilized the method to process newspapers, periodicals, data reports, etc. Our text recognition methods have evolved from traditional text recognition methods to mainstream deep learning text recognition methods [13,14] and multimodal text recognition methods in more cutting-edge disciplines [15,16] from the 1970s to the present. Text detection and recognition consist of three primary tasks: text detection, text recognition, and end-to-end recognition. In terms of text irregularities in visual images, text distortion, multiple scales, text recognition accuracy, and speed, etc., exist. The dominant deep learning text recognition methods have made significant progress. However, as the requirements for text recognition tasks become more diverse, emerging multimodal text recognition methods become more difficult. The traditional text recognition method employs a manual feature extraction technique, which is simple to comprehend but lacks generalizability. The deep learning text recognition method uses a convolutional neural network method which improves the neural network text recognition results through deep learning attention. This improves the accuracy and robustness of text recognition, making it better than the traditional text recognition method. With technological developments in

single-modal AI technology, there is a developing desire among academics for multimodal information technology. Methods for multimodal text recognition attempt to combine visual feature information such as image, video, audio, etc., first through translation [17] and alignment [18] of modal information. The model is then pretrained to facilitate cross-modal interaction between features to learn about the concealed wealth of information between modalities. Finally, the model can accurately extract the information of characters and fields, and it has a more pronounced effect enhancement in generalization ability and scene adaptation, making it superior in scene application to the deep learning text recognition method. For multimodal text recognition, there are still many issues that require further investigation. This chapter examines the deficiencies of the text detection and recognition modules in multimodal scenes. In the detection module, we proposed the LK-PAN network of large receptive fields to improve the CML distillation strategy and then employed an RSE-FPN with a residual attention mechanism. Experimental verification has shown that it can perform well in detecting large font text and extreme aspect ratio text in image and video datasets. In the recognition module, we modified the original CTC decoder to use the GTC method and applied a sinusoidal loss function for rotation recognition. This loss function can optimize the model's recognition of rotating fonts and improve text detection performance in settings with arbitrary rotation angles. Following validation on image and video datasets, the detection and recognition performance of arbitrarily rotated text has been enhanced to some degree. In two sections, Sections 2.1.1 and 2.1.2, the research related to text detection and recognition is elucidated.

### 2.1.1. Text Detection

Typically, traditional detection methods [19] and deep learning detection methods [20,21] are classified as detection methods in the field of text detection. First, the original data need to be preprocessed, which includes data segmentation, noise reduction, correction, and compression, among others. The preprocessing step can effectively resolve issues with uneven illumination, blurry handwriting, and the removal of sensitive information. The majority of traditional text detection methods rely on manual design features to capture the target text attributes. However, the majority of them are unable to solve for the effects of image blurring [22] and text distortion [23] on the detection results. In order to effectively solve the aforementioned constraints, deep learning methods extract useful features from training data. Regression-based methods and segmentation-based methods classify the majority of deep learning text detection techniques [24]. For regression-based detection methods, academics typically make use of widely used networks, such as Faster R-CNN [25], YOLO [26], CTPN [27], TextBoxes [28], and PCR [29]. However, these detection methods are ineffective at detecting text sequences that are infinitely long, and it can be difficult to obtain text wrap-around curves that are smooth. In order to solve that problem, academics have proposed image segmentation-based text detection algorithms. The segmentation-based detection method regards the detection problem as a text classification problem by predicting text instances at the pixel level and then detecting multiangle, irregular text using the concept of semantic segmentation. Mainstream network topologies primarily consist of PSENet [30], DB [31], and FCENet [32], which are superior to regression-based detection methods. Other academics have proposed combined regression and segmentation detection methods, which would make it easier to find irregular text, reduce false alarms, and make text more robust at different scales, among other things. The main network models proposed by researchers are PMTD [33], etc. The accuracy of detection is better than that of the segmentation detection method, but the performance of text detection in sequences with arbitrary rotation angles needs improvement. In the text detection module, we apply this to solve the problem of a lack of detection models for big font text or text with an extreme aspect ratio in existing multimodal scenes.

### 2.1.2. Text Recognition

Text recognition is the task of recognizing text in the context of unregulated scenarios. Earlier methods relied on character or word recognition but were unable to recognize characters and phrases in texts of arbitrary length and dictionaries. Since the deep learning text recognition method was proposed by academia, it has effectively solved the above disadvantages. For example, ShiB et al. proposed a CRNN network model based on structured sequence recognition [34], and YinF et al. proposed a sliding window SlidingCNN-based method [35]. Text recognition methods based on deep learning can be classified in general as connectionist temporal classification (CTC)-based methods, attention-based methods, and end-to-end methods. The CTC mechanism accumulates conditional probabilities during the prediction phase and converts the neural network output features into string sequences to successfully solve the problem of temporal text alignment [36]. The most prevalent network models are DTRN [37] and others. However, these recognition methods are ineffective for finding character correlations. In order to obtain more important information, academics have proposed attention-based methods such as single-headed attention, multiheaded attention, and masked multiheaded attention. These methods have a greater improvement in solving intercharacter association information, attentional bias, and fuzzy text recognition. Common networks include MORAN [38] and ASTER [39]. In recent years, a number of academics have also proposed combining CTC and attention to improve the accuracy of text recognition; for example, SCATTER [40] is one such method. The end-to-end text recognition method is able to tightly couple the detection and recognition modules [41], which improves the model volume, detection speed, and recognition accuracy compared to the previous two phases of text recognition. In the text recognition module, text recognition for any angle of rotation is one of the study problems covered in this paper.

### 2.2. Text Sentiment Analysis

Text sentiment analysis, also known as opinion mining, is a concept that was developed in 2003 by Nasukawa et al. [42]. Its primary focus is on learning how to extract subjective information from textual content and how to perform near-human reasoning in order to represent emotions, views, or feelings. The application areas it investigates include internet opinion monitoring [43], analysis comments, business investment opportunities, etc. Existing methods for sentiment analysis can be divided into multiple groupings, including multimodal sentiment analysis, methods that are based on machine learning, methods that are based on deep learning, and methods that use sentiment lexicons. This method that is based on a sentiment lexicon is primarily dependent on the development of a sentiment lexicon, which initially extracts sentiment words and then performs sentiment evaluation on the information that was taken from the words based on the sentiment lexicon. The How Net [44] and NTUSD [45] lexicons are two examples of popular generic sentiment dictionaries. Academics have made use of machine learning techniques to improve sentiment analysis. Two examples of these algorithms are the support vector machine (SVM) [46] and the Nave Bayesian (NB) [47]. Machine learning is a relatively new field of study. On a general test set, the accuracy of sentiment discrimination was found to be superior to that of the lexicon approach. Nevertheless, the success of the approach is dependent on the pre-selection of features, and making an excellent feature engineering selection is a laborious and time-consuming process. Most deep learning methods use convolutional neural networks to extract text features, learn the relationship between features in many ways to fully explore hidden feature information, and then classify after the algorithm is processed: for example, MCCNN [48], etc. Deep learning techniques are superior to older methods in terms of their ability to represent data and generalize models. However, this improvement comes at the cost of increased model complexity and an inability to be interpreted. Over the course of the past few years, multimodal sentiment analysis has emerged as a topic of intense attention among academics. The paper introduces a new model for sentiment analysis and makes improvements to it in order to

increase the model's accuracy in monitoring the positive and negative information of the internet's public opinion.

### 2.3. Multimodal Visual Text Recognition

The growth of media information, primarily in the form of images and videos, has been made mainstream by the ongoing development of multimedia and communication technologies. As a result, the question of how to extract textual information from multimodal data has emerged as an important area of investigation in recent years. For example, Zhang et al. proposed a cross-modal depth metric learning-based oracle character recognition method to address the problem that it is difficult to obtain topical oracle character samples in oracle character images. They achieved cross-modal recognition of topical oracle characters by modeling the shared feature space and nearest neighbor classification of copied and topical oracle characters. This method was successful in solving the problem [49]. ActBERT is a multimodal video and text self-supervised learning model that was proposed by Zhu et al. for a visual center that lacks the ability to adequately explain both local and global visual content at the same time. The new model solves these problems by incorporating global action and local area features in the input layer, as well as designing a new coding module for multimodal feature learning. In addition, the output layer of the new model includes a new coding module. The findings of the experiments indicate that the ActBERT model shows powerful learning potential on large-scale datasets [50]. Gao et al. solved the problem of needing to read text in images in question-and-answer tasks, which poses a challenge to existing models, with the most common difficulty being the frequent presence of rare words and polysemantic words in pictures. To solve these problems, the authors designed a new VQA model, MM-GNN, which exploits the rich information of multiple modalities in images to infer the semantics of text in images, and experimental results show that it can help downstream tasks well [51].

### 3. Methodology

In the following chapters, we will provide a comprehensive explanation of the multimodal text recognition and sentiment analysis model (MTR-SAM) that we have proposed. In this paper, we solve the problem of insufficient text detection models in existing multimodal scenes for large font text or text with an extreme aspect ratio. We solve the issue of low accuracy of text recognition models for rotating fonts in existing multimodal scene font text. Secondly, we solve the technical problem that existing sentiment analysis models are unable to well solve the problem of multiple meanings of the word text sequences and the information loss of intermediate vectors during feature encoding, which will lead to a lack of contextualization of the modeled features. Thus, to solve the above issues and combine data characteristics in opinion analysis, we will first improve and modify the OpenVINO-deployed PP-OCR model based on the latest text recognition advances, and then we will improve and modify the sentiment analysis model. Finally, the model for better text recognition and the model for analyzing sentiment are combined, and then a new model for multimodal text recognition and sentiment analysis is designed. The work is manifested in three areas: (1) The LK-PAN network with large sensory fields is proposed in the detection module to upgrade the CML distillation strategy, followed by an RSE-FPN with a residual attention mechanism to enhance the characterization of the feature maps. By improving the text detection module, we can address the problem of insufficient text detection models in existing multimodal scenes for large font text or text with an extreme aspect ratio. (2) The original CTC decoder is changed to the GTC (Guided Training of CTC) method to further enhance the inference of the model for multimodal text recognition accuracy. A sinusoidal loss function for rotation recognition is also introduced, which can optimize the model's recognition of rotated fonts and enhance the performance of text detection for scenes with arbitrary rotation angles. By improving the text recognition module, we can solve the issue of the low accuracy of text recognition models for rotating fonts in existing multimodal scenes. (3) We propose the introduction of a sentiment analysis

model for distinguishing multimodal recognition results. First, the BERT model is adopted to acquire high-quality semantic encoding to solve the problem of a single word having multiple meanings. Second, a combination of global and local information block attention mechanisms is proposed as a solution to the problem that word vectors readily lose word positions during the splicing process and can result in information loss in intermediate vectors during encoding. The proposed model can assist public opinion managers with intelligent monitoring. The structure diagram of the improved model is shown in Figure 1.

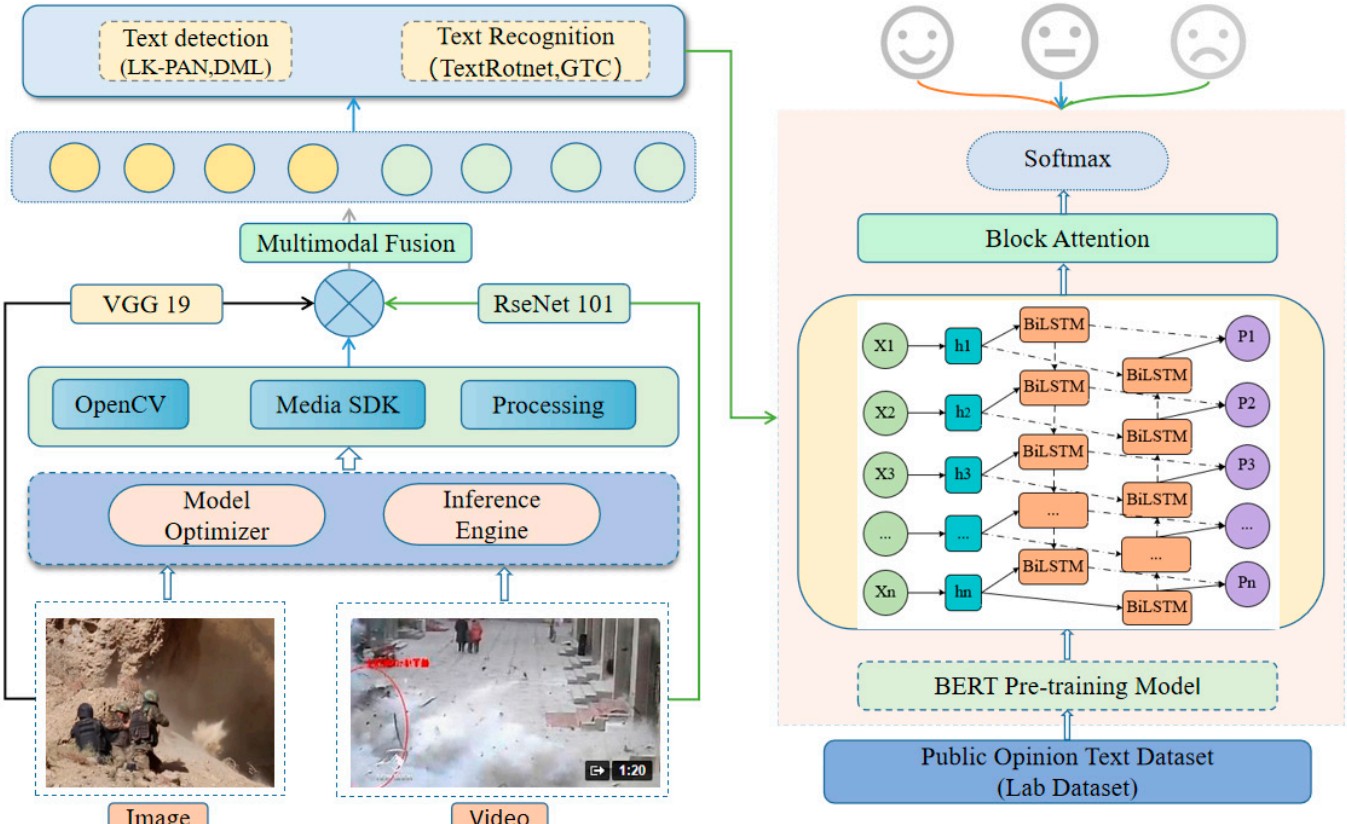

**Figure 1.** Multimodal text recognition and sentiment analysis model MTR-SAM.

In Figure 1 above, the datasets of images and videos are prepared in the first stage as the input for the training model. Second, after the second stage of the inference acceleration engine, which mainly includes two parts, the model optimizer and inference engine, can accelerate the reading and inference of input images and videos by using text detection and recognition models. Then, in the third stage, the three modules OpenCV, Media SDK, and Processing optimize the input information (image and video data) from the previous phase in order to improve the image and video quality. Immediately thereafter, the image and video data are put through feature extraction by the VGG 19 and RseNet 101 networks, respectively, and the extracted features are mutually combined with the third-stage output information as the subsequent input. In the fourth-stage text detection module, the LK-PAN and DML strategies are added, and in the text recognition module, the TextRotNet network and GTC strategies are added. Then, the improved sentiment analysis model is pretrained on the self-built opinion dataset in this paper. The primary procedure of the sentiment pretraining model is as follows: The preprocessed dataset is first used as input to the BERT model [52], which is capable of producing deep bidirectional linguistic representations from text input sequences. Second, after the BERT model, the output is put into the BiLSTM network to acquire additional multifeatured textual context information. Then, following the enhancement of the block attenuation mechanism and normalization by the softmax function, a new model of sentiment analysis for public opinion monitoring

can be accomplished. Finally, the combined multimodal text recognition and sentiment analysis model is effective at detecting the polarity of sentiment in risky images (videos). The experimental results also show that the improved model has a greater enhancement on the dataset than the original model.

### 3.1. Improvement Strategy in Text Detection Module

This thesis upgrades the CML distillation strategy of the original PP-OCR. It combines the standard distillation of the traditional teacher guiding students with the DML (Deep Mutual Learning) between the network of students and the teacher network guiding the students while they are learning from each other. In this paper, the LK-PAN of the PAN module with a large feeling field is used for the optimization of the teacher model. The LK-PAN framework diagram is shown in Figure 2. First, the input feature information is extracted by ResNet50 network features, which are a stack of multiple similarly structured blocks that are the basic units of the residual network, residual blocks. This network can accelerate the training speed and improve the training effect of the model. The gradient and feature degradation problems can be well solved when the model is deepened with layers to obtain rich features. Second, after the LK-PAN module, the core is to increase the convolutional kernel in the path augmentation (PAN) structure, and the size of the convolutional kernel is expanded from 3 × 3 to 9 × 9. By increasing the size of the convolutional kernel, the perceptual field covered by each position in the feature map is enhanced. In image and video datasets, it can show good results for detecting text with large fonts and text with extreme aspect ratios while combining the LK_PAN network with the DML distillation strategy. Finally, the information that passes through the LK-PAN network is concatenated.

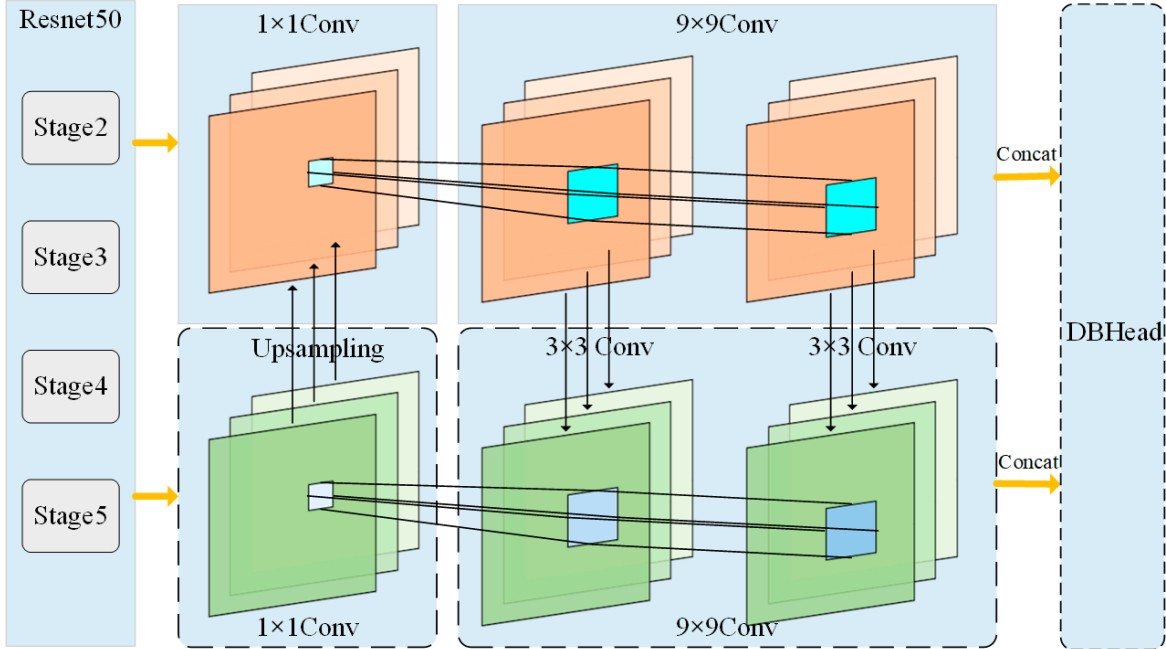

**Figure 2.** LK-PAN model framework.

For the student model, an RSE-FPN (Residual Squeeze-and-Excitation Feature Pyramid Network) with a residual attention mechanism is used in this paper [53]. Figure 3 illustrates the RSE-FPN framework structure. First, the feature information is passed through the MobileNetV3 network, which automatically acquires the importance of each feature channel through learning, and the results are used to enhance useful features and suppress features that are less useful for the task at hand. Then, the feature information from the previous step is used as input for the next layer of the RSEConv network. This network improves the convolutional layer in FPN into a channel attention structure RSEConv layer with residual

structure by introducing a residual structure, which has better characterization ability for feature maps. Finally, the feature information that passes through the RSEConv network is connected.

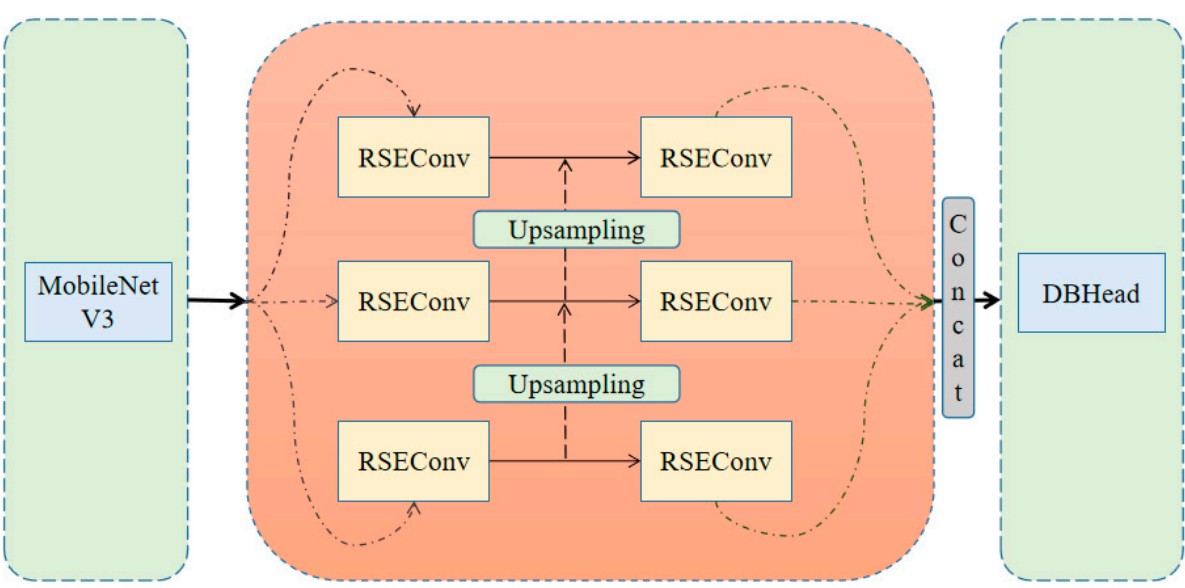

**Figure 3.** RSE-FPN framework structure diagram.

### 3.2. Improvement Strategy in the Identification Module

Text recognition of multimodal data such as contorted deformation and tilt with direction is inadequately supported by the majority of methods. For example, in the recognition task, the accuracy of the extracted text after reversal or rotation is low when the image contains text with a directional inclination. Even if it is possible to recognize the image region containing the text, it is difficult to extract that text from the image. Therefore, in the context of public opinion monitoring, it is very meaningful to predict the direction of text from multimodal data such as images or videos. In this paper, we introduce a sinusoidal loss function for rotation recognition proposed by Roi Ronen [54], which can optimize the model for recognition of rotated fonts and enhance the performance of text detection for scenes with arbitrary rotation angles. The specific functional designs are as follows:

For target rotated text area detection $r \in R^{N*5}$, a new direction-aware loss function $L_{rbox}$ is proposed. The loss function sets five description parameters, two of which describe the coordinates of the center of the rotated region; two are used to describe the height and width of the detected region; and the last one is used to describe the angle of rotation. $\theta \in R$, $\theta$ indicates the angle toward the top. The loss function for the first m match detection is expressed as shown in Equation (1):

$$L_{rbox} = \sum_{i=1}^{4} \alpha_i |\hat{r}_i - r_i| + \alpha_\theta \sin^2(\hat{\theta} - \theta) \tag{1}$$

where the hat indicates a prediction. The values of both $\alpha_i$ ($i$ takes values in the range [1, 4]) and $\alpha_\theta$ are chosen empirically. For $n \in Z$, the sine square loss function has $n\pi$-periodicity, and this loss function takes advantage of the fact that the symmetry between the rectangular frame in the rotated region and m eliminates the preexisting ambiguity during model training and is able to make the same prediction for a vertical or inverted detection frame. At the same time, L1 parametric regularization is used for the first four parameters and a loss based on the sine function is designed for the angle, which satisfies reasonableness and derivability. After introducing this loss function on top of the original

model, it is verified on the test dataset that the performance of detection and recognition of arbitrary rotation text sensitivity has improved to some extent.

The original CTC decoder has the ability to make rapid inferences, but its accuracy is poor. Thus, it introduces the GTC method. The attention module is used to guide the CTC training to integrate more features, while it is removed without increasing the inference time during the inference phase. Attention is more attuned to spatial data and can optimize STN (Spatial Transformer Network) networks with precision [55].

### 3.3. Sentiment Analysis Model Improvement

Existing datasets of internet public opinion texts (about terrorism, violence, etc.) are small, which means that trained models will not be very general. In addition, existing text sentiment analysis methods depend on how well words are separated and cannot solve the problem of a single word having more than one meaning. The paper proposes a text sentiment analysis model for internet public opinion data. First, the lab dataset, a Mandarin text dataset of internet public opinion, is automatically constructed through the use of web crawler technology. It contains five types of data: 3211 in the category of terrorism, 2864 in the category of violence, 3102 in the category of gambling, 3512 in the category of pornography, and 6453 in the category of normal. Next, the dataset is annotated using the BIO annotation method. Finally, a sentiment analysis model that makes use of BERT and a block attention mechanism is put through its paces to pretrain the internet public opinion dataset. The flowchart of the sentiment analysis model is shown in Figure 4:

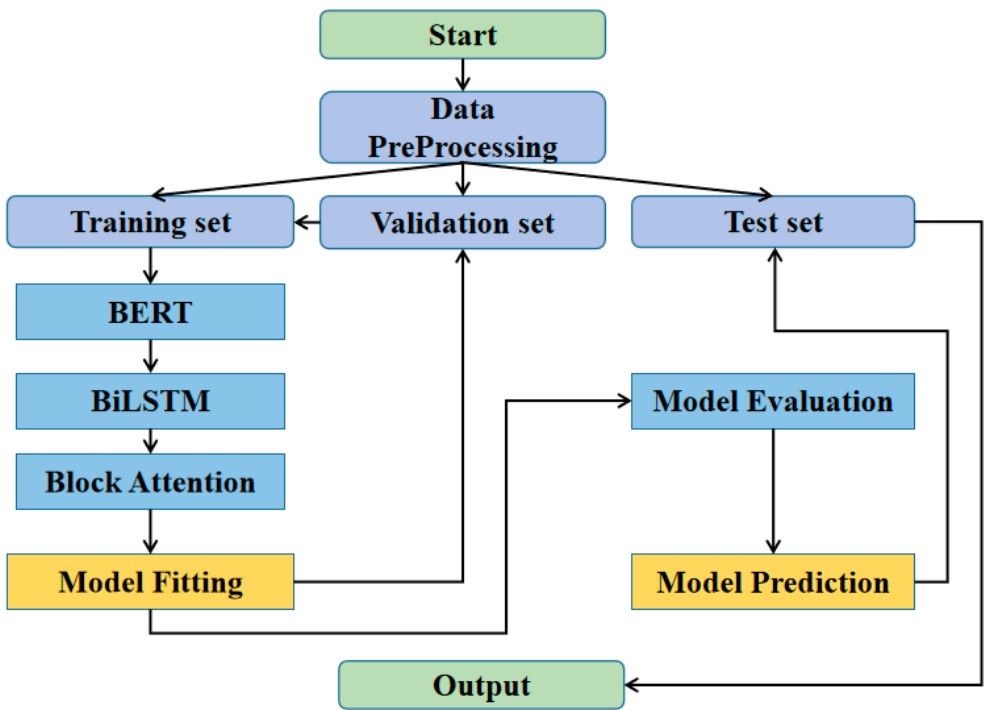

**Figure 4.** Overall flow chart of the sentiment analysis model.

In Figure 4 above, the crawler is first used to capture internet public opinion information in the first stage (start). Second, after the second stage of data preprocessing, we obtain the self-built internet public opinion dataset Lab Dataset (including five kinds of labeled data such as those related to terrorism and violence) and divide the constructed dataset into a training set, validation set, and test set at a ratio of 8:1:1. Next, in the third stage (shown on the left side of Figure 4), the data from the preprocessed training set are put into the BERT model, which is able to generate deep bidirectional language representations based on text input sequences. This process takes place on the left side of the figure. Following this, the vector features are used as the next input, and a BiLSTM network is utilized in order to

further gather multifeature textual contextual information. Finally, the improved block attention mechanism (block attention stage), which enhances the contextual relationship between words, is able to filter the features of the deformed sentence matrix. The model fitting stage evaluates the variance of the model estimates and identifies whether there are any overfitting problems in the model fit. If there are overfitting or underfitting problems, the validation set and training set are returned and retrained until the model fit is up to standard. If there is no fitting problem, we enter the model evaluation phase, which mainly ensures that the BN (batch normalization) layer can use the mean and variance of all training data and the accuracy, recall, and F1 values as the model evaluation index. The next phase (model prediction) validates the experimental effect on the test set and finally outputs the sentiment polarity discrimination results.

### 3.3.1. BERT

Figure 5 shows the processing data structure of the BERT model (the model pretraining layer), in which cleansing data from the second stage (data preprocessing) are first transmitted into the third stage (left side of Figure 4). The preprocessed text K will obtain three vector representations: token embedding, position embedding, and segment embedding. These vector representations will be added element-by-element to generate the embedded segment $\{[cls], Tok1, Tok2, \ldots, TokN\}$. In turn, it is used as input to the BERT encoding layer and is packaged as $\{E_{cls}, E_1, E_2, \ldots, E_n\}$. Second, the input sequence is pretrained by L-Transformer network layers to refine the token-level features layer by layer. The final word vector $T^{n*m} = \{T_{cls}, T_1, T_2, \ldots, T_n\}$, containing the contextual semantics, is obtained, where the dimension of the word vector is defined as m and the sentence length is defined as $n$.

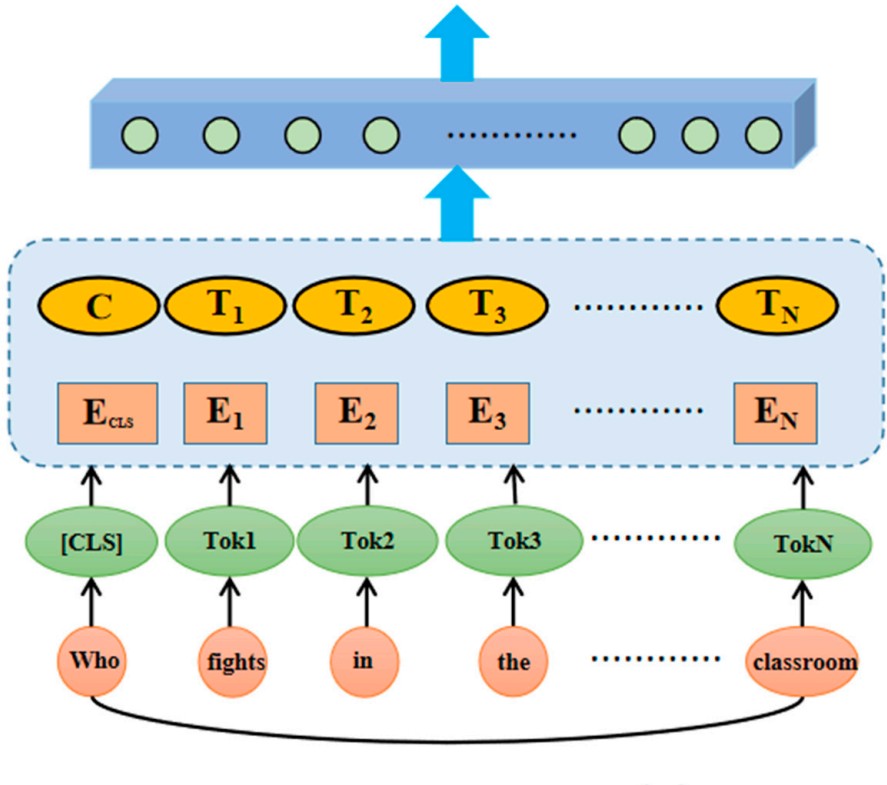

**Figure 5.** The structure of BERT.

The algorithm for calculating this model is shown in Equation (2), where *Q* and *K* in Equation (2) are the feature vectors for calculating the attention weight, *V* is the input

feature vector, and *Attention*() is the corresponding weight multiplied by *V* according to the degree of attention.

$$head_i = Attention(QW_q^i, KW_k^i, VW_v^i) \tag{2}$$

Equation (3) is calculated by first calculating a single attention mechanism $head_i$ ($i \in (1, n)$). Second, multiple self-attentive mechanisms are computed in parallel, and then, the output is obtained by multiplying the randomly initialized matrix *W* by *n* groups and finally by a linear transformation.

$$Multihead = concat(head_1, head_2, \dots, head_n)W \tag{3}$$

The model separates the text in the opinion dataset word by word and generates dynamic vectors, which not only avoids ambiguity caused by word separation but also solves the problem of a word having multiple meanings by combining it with its contextual sense.

### 3.3.2. Attention Layer

The primary deficiencies of the present attention mechanism are as follows: (1) During the splicing process, the word vectors lose word positions with relative ease. (2) It results in the loss of information from intermediate vectors during the encoding process, which may influence the semantic richness. Therefore, in order to solve these problems, in this paper, the block attention mechanism is used in the third stage (left side of Figure 4), which enhances the contextual relationship between words and can filter the features of the deformed sentence matrix, resulting in enhanced attention to the features associated with the emotional polarity of the text. The original block attention mechanism only considers global information and does not consider the local relationship representation information between words. This scheme improves the original block attention mechanism strategy by adding the local association features of words, enhancing the importance of acquiring each channel of the feature map, and broadening the information captured by the network model. The exact calculation process is as follows:

The weight values $y_1$ and $y_2$ of the word block matrix $x_i$ are obtained by fusing the global and local information, respectively, and the Equations are shown in (4) and (5):

$$y_1 = \frac{1}{1 + e^{(-(W_1 * x_i + b_1))}} \tag{4}$$

$$y_2 = \frac{1}{1 + e^{(-(W_2 * x_i + b_2))}} \tag{5}$$

In Equations (4) and (5), sigmoid denotes the activation function that can provide nonlinear modeling capability for the attention mechanism to map the output results to the range [0, 1]. $W_1$ and $W_2$ denote the weight matrix, and $b_1$ and $b_2$ denote the bias.

$$h_i = [y_1 * x_i \,;\, y_2 * x_i] \tag{6}$$

Immediately afterwards, the weight matrices $y_1$ and $y_2$ calculated by Equations (4) and (5) continue to be multiplied with the word block matrix xi and then spliced to obtain the feature vector matrix $h_i$ after information attention (Equation (6)), and the final output contains the vector matrix $M^{FE}$ of hi after the block attention mechanism, where $M \in R^{2d_h}$ and dh are the dimensions of the contextual embedding vectors. The improved block attention mechanism is shown in Figure 6.

Figure 6 fuses the global and local feature information and finally splices the two feature information to broaden the model to capture the feature information of the context, thus improving the model's accuracy. Immediately following Equation (6), the output features $M^{FE}$, obtained from the attention layer, are first input to a multilayer perceptron,

and finally, the emotional polarity is predicted using the softmax layer, which is calculated as shown in Equations (7) and (8):

$$L = MLP(M^{FE}) \tag{7}$$

$$V = softmax(W_p L + b_p) \tag{8}$$

The *MLP* in Equation (7) represents the multilayer perceptron. *P* in Equation (8) has the same dimensions as the sentiment labels in the opinion dataset. $W_p \in R^{d_h}$ denotes the learning weight matrix, and $b_p \in R^{d_h}$ indicates the bias.

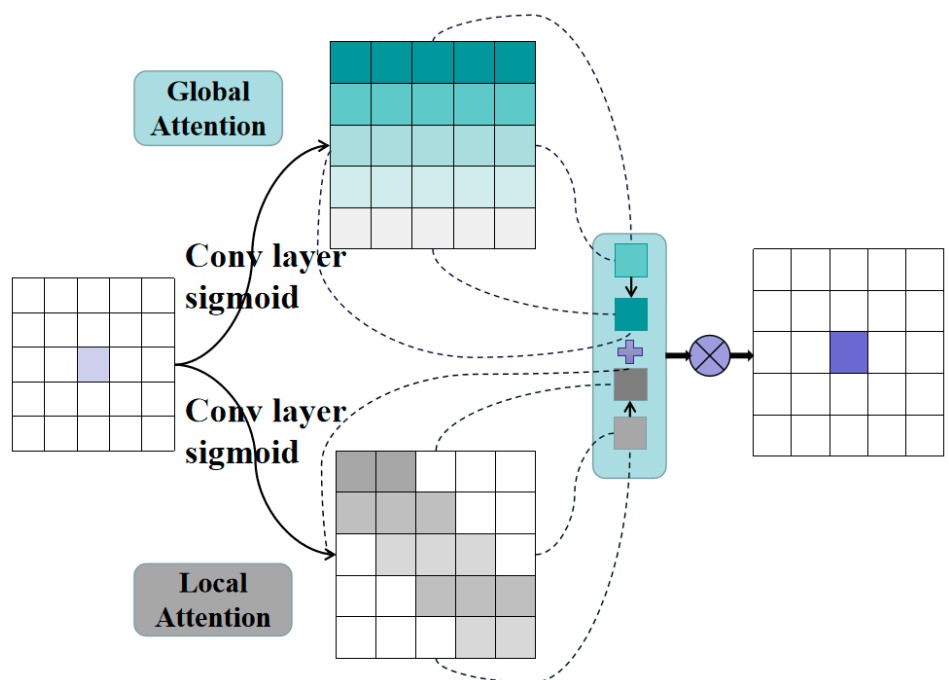

**Figure 6.** Improved block attention mechanism model diagram.

## 4. Systematic Experimental Process and Results Analysis

### 4.1. Experimental Environment

This experiment is implemented in the Windows 10 environment (Microsoft, Redmond, WA, USA). The hardware configuration of the experiment is a central processing unit (CPU) using an Intel(R) Core(TM) i5-7200U CPU @ 2.50 GHz (Intel, Santa Clara, CA, USA) and 12 GB of RAM. The graphics processor (GPU) is an NVIDIA Tesla V100 (Nvidia, Santa Clara, CA, USA) with 24 GB of RAM; the framework used is PyTorch; and Python (Python, Fredericksburg, VI, USA) is used for all of the models in this paper.

### 4.2. Experimental Dataset

In this thesis, the open-source ICDAR 2015 dataset [31] and video data collected by crawlers are used in multimodal text recognition experiments to validate the model for text recognition on images and videos, respectively. On the image dataset, the training sample comprises 1000 randomly selected images from the ICDAR 2015 dataset, and the test sample comprises 500 randomly selected images. The majority of the images contain English text. On the video dataset, 101 opinion videos were crawled from the internet using crawler technology, with the shortest video clocking in at 11 min and the longest clocking in at 23 min. The majority of the text in the videos is in Chinese (Mandarin) and English. Adobe Premiere Pro (Adobe, San Jose, CA, USA) was also utilized to crop these 101 original captured videos in order to complete the video dataset. In this paper, we crop and select 3261 short video clips with opinion information text from 101 videos (each short video is

approximately 25 s long) and divide the 3261 short videos into training sets, testing sets, and validation sets in the ratio of 8:1:1. There are 2608 short video clips in the training set, 326 short video clips in the testing set, and 327 short video clips in the validation set. We perform semi-automatic annotation on the 3261 short video clips. Using a training image in ICDAR 2015 as an example, the original image and data annotation format are shown in Figure 7 and Table 1:

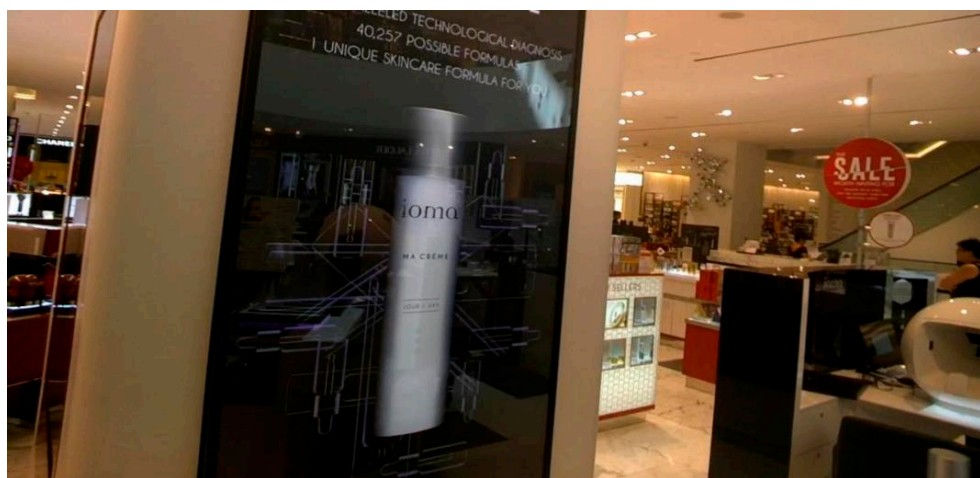

**Figure 7.** A sample image from ICDAR 2015.

**Table 1.** A sample image in ICDAR 2015 annotated with partial text content of the format (corresponding to Figure 7).

| Single Test Image | Marked Out Text | Coordinate Points |
|---|---|---|
| | "FORMULA" | [[570, 81], [645, 101], [644, 121], [568, 101]] |
| | "DIAGNOSIS" | [[654, 42], [730, 62], [727, 82], [652, 62]] |
| | "40,257" | [[456, 17], [521, 34], [515, 57], [451, 41]] |
| img_891.jpg | "POSSIBLE" | [[520, 34], [597, 52], [594, 77], [516, 60]] |
| | "TECHNOLOGICAL" | [[524, 1], [661, 42], [660, 63], [523, 22]] |
| | "FORMULAS" | [[596, 54], [675, 75], [673, 97], [594, 76]] |
| | "UNIQUE" | [[408, 38], [491, 60], [487, 85], [405, 64]] |
| | "FOR" | [[645, 100], [677, 107], [675, 126], [643, 118]] |
| | ... | ... |

First, the BIO annotation method is used to annotate the dataset, and the annotation process is used to annotate the beginning and end tokens as "B" and "O", respectively, and the remaining tokens as "I", as shown in Figure 8. Second, the pretraining data of the improved sentiment analysis model, the Lab Dataset, an opinion text dataset automatically constructed by web crawler technology, are shown in Figure 9. It includes 3211 terror-related categories, 2864 violence-related categories, 3102 gambling-related categories, 3512 porn-related categories, and 6453 normal categories, which are five kinds of negative labels (terror-related, violence-related, gambling-related, and porn-related) and normal categories (both positive and neutral labels).

| Sentences | : | | In fact, you are too reckless to fight with evil people | | | | | | | | | |
|---|---|---|---|---|---|---|---|---|---|---|---|---|---|
| Sentences | : | 其 | 实 | 你 | 和 | 恶 | 人 | 打 | 架 | 太 | 鲁 | 莽 | 了 |
| Tags | : | O | O | B-People | O | B-People | I-People | I-Behavior | I-Behavior | O | B-Behavior | I-Behavior | O |

**Figure 8.** A sample text sequence BIO sample annotation sample.

**Public Opinion Dataset Lab Dataset**

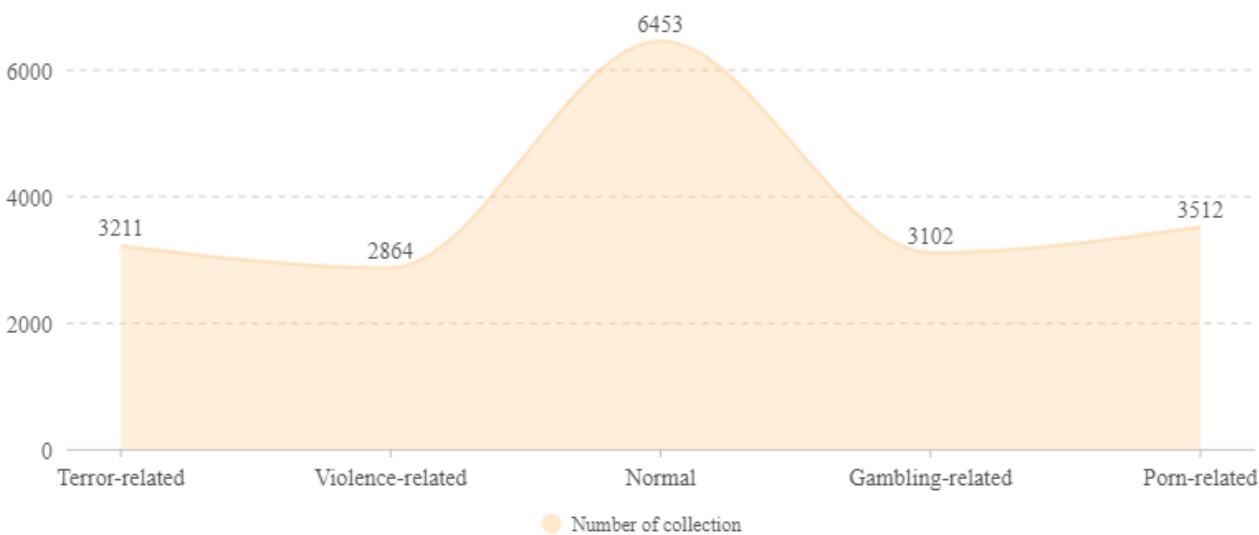

**Figure 9.** Public opinion lab dataset.

*4.3. Evaluation Metrics*

The evaluation metrics for this experiment are accuracy *P*, recall *R*, and *F*1 value. The accuracy rate represents the ratio of the number of samples correctly classified to the total number of samples on the test dataset. Recall represents the ratio of the number of correct predictions to the number of all correct predictions in the test dataset. The *F*1 value represents the summed average between *P* and *R*. The formulae are shown in Equations (9)–(11):

$$P = N'/N \tag{9}$$

$$R = N'/M \tag{10}$$

$$F1 = \frac{(\alpha * \alpha + 1)}{(P + R)\alpha * \alpha} P * R \tag{11}$$

where $N'$ denotes the number of sentiment words correctly identified by the model, and $N$ and $M$ denote the number of sentiment words that are all predicted to be correct and the number of sentiment words that are all actually correct, respectively. The recall rate has more influence on the *F*1 value when $\alpha > 1$ and the accuracy rate when $0 < \alpha < 1$.

*4.4. Experimental Results*

4.4.1. Comparison of Experimental Results of Multimodal Text Recognition

Table 2 shows the comparison of the detection speed effect of the network model on the video dataset before and after improvement. From Table 2, it can be seen that compared to the model before improvement, the average recognition speed of the model for text in video increased from 19.2 frames per second before to 25.3 frames per second, an improvement of 31.77%.

**Table 2.** Comparison of the effect of model detection speed before and after improvement.

| Test Dataset | Network Model | Average Detection Speed (Frames/s) |
|---|---|---|
| Video data collected by crawler | Model before improvement | 19.2 |
| Video data collected by crawler | Improved model | 25.3 |

Figure 10 shows the comparison of the accuracy, recall, and F1 value effects of the network model on the image and video datasets before and after the improvement. As shown in Figure 10, compared to the model before improvement, the detection effect is slightly improved after adding the GTC method and the sinusoidal loss function scheme for rotation recognition. The *F*1 value of the text detection effect on the image dataset ICDAR 2015 is improved by 12.58%, and the *F*1 value of the text detection effect in the video is improved by 10.78%.

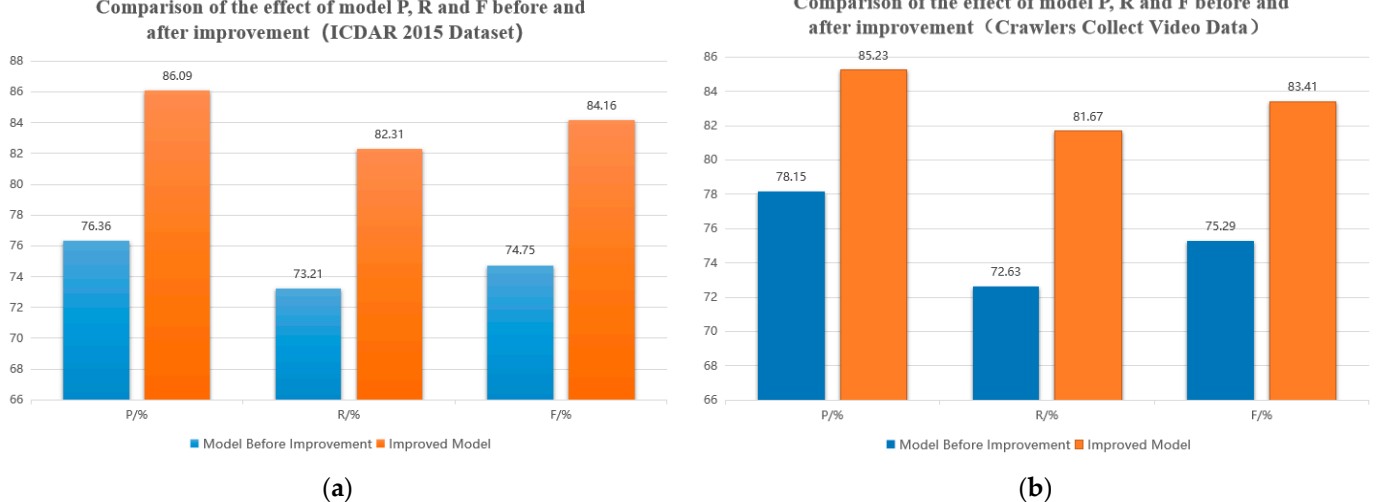

(**a**)                                                                                  (**b**)

**Figure 10.** Comparison of the effect of models P, R and F before and after improvement; (**a**) ICDAR 2015 dataset test results; (**b**) Test results on the video data collected by the crawler.

Taking a single image from the ICDAR 2015 dataset, Figure 7, as an example, Table 3 shows the comparison of the text detection effect in the image before and after the improvement. The results in Table 3 show that the improved model can not only recognize more "MACREME" text but also improve the confidence of recognition because the improved model is slightly slower than the premodified model in terms of detection time. Therefore, the recognition ability of the improved model has improved.

**Table 3.** Comparison of image text detection effect before and after improvement.

| Network Model | Image Text Detection Effect | Detection Confidence |
|---|---|---|
| Model before improvement | 40.257POSSIBLE FORMULA | 0.815 |
| | UNIQUE SKINCARE FORMULA FOR | 0.936 |
| | SALE | 0.913 |
| | omo | 0.738 |
| Improved model | 40.257POSSIBLEFORMULA | 0.837 |
| | INOLOCICALDIACNOSIS | 0.859 |
| | UNIQUESKINCAREFORMULAFOR | 0.949 |
| | SALE | 0.927 |
| | lomd | 0.695 |
| | MACREME | 0.924 |

### 4.4.2. Comparison of the Experimental Results of the Emotion Analysis Model

The data source can strengthen the model's adaptation performance, and the quality of the model also determines the quality of the data analysis. Therefore, to verify the effectiveness of the models proposed in this paper, the following comparison is made with commonly used models:

- GRU [56]: This model has one less internal "gating" mechanism than LSTM, but the experimental results are comparable to LSTM.

- BiLSTM [57]: The model was proposed by G. Xu et al. to better capture the semantic dependencies in both directions.
- TextCNN [58]: B. Guo et al. did not introduce an attention mechanism in this model and did not consider contextual relationships.
- BERT-BiGRU [59]: A BERT-BiGRU-CRF entity recognition model is proposed for the text data of electronic cases, which can obtain the best sequence information, but the potential information of electronic medical records needs to be further explored.
- BERT-BiLSTM [60]: In the process of information extraction, the model is capable of F1 values up to 75%, but not by considering the important parts of the many extracted pieces of information.
- BERT-BiLSTM + Attention [61]: The model uses BERT as a word embedding and combines BiLSTM with attention to achieve the desired inference.

In Figure 11, the performance of machine learning-based sentiment analysis methods is slightly inferior to that of deep learning in the experimental effect of sentiment analysis due to the complexity of manually produced features and the human subjective factors that can influence prediction results. Existing sentiment analysis models based on BERT that have been enhanced on the basis of BERT achieve great outcomes, but there is a significant disadvantage in that the improved method considers all words in a sentence and cannot center in on the most significant parts of the text. This paper's proposed method of combining BERT with a block attention mechanism takes into account the benefits of both local and global contextual features, and the P value, R value, and F value of the experimental effect are all greater than those of other baseline models.

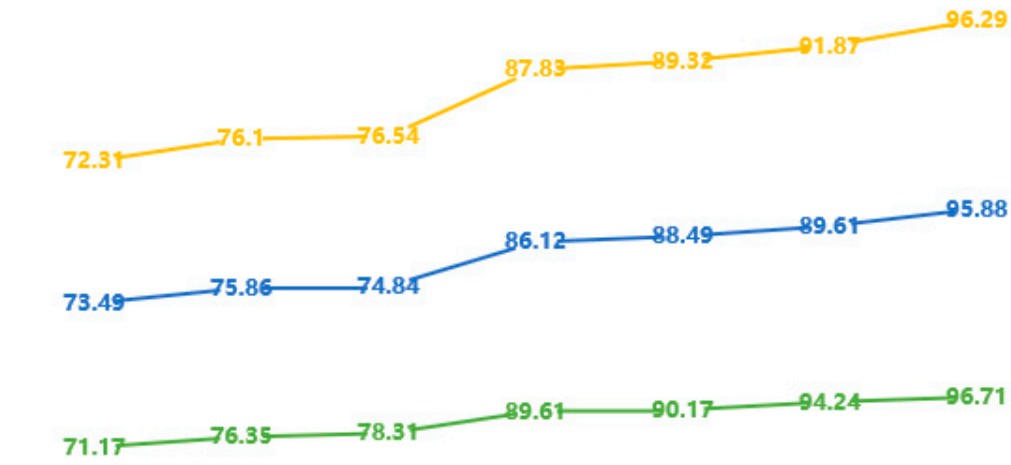

| | GRU | BiLSTM | TextCNN | BERT-BiGRU | BERT-BiLSTM | BERT-BiLSTM-Attention | Improved Model |
|---|---|---|---|---|---|---|---|
| F/% | 72.31 | 76.1 | 76.54 | 87.83 | 89.32 | 91.87 | 96.29 |
| R/% | 73.49 | 75.86 | 74.84 | 86.12 | 88.49 | 89.61 | 95.88 |
| P/% | 71.17 | 76.35 | 78.31 | 89.61 | 90.17 | 94.24 | 96.71 |

**Figure 11.** Commonly used model comparison tests.

The data in Table 4 correspond to Figure 11, with the P value corresponding to the green line, the R value to the blue line, and the F value to the yellow line. In addition, Figure 12 shows a comparison and analysis of the *F*1 value updates for the original 25 cycles. In the traditional neural network model, the *F*1 value begins training at a low level, reaches a high level after multiple iterations, and then stabilizes. The F1 values of the three models fused with BERT are higher than the *F*1 values of the traditional neural network models at the beginning of training, and the F1 values continue to increase and trend toward a higher value. In this paper, the *F*1 value of the improved model is comparable to that of the three BERT fusion models at the start of training; however, after iterations, the *F*1 value surpasses that of the fusion model and tends to become stable, demonstrating that the improved model has an improvement effect.

**Table 4.** Comparison test results of commonly used models (Corresponding to Figure 11.).

|  | GRU | BiLSTM | TextCNN | BERT-GRU | BERT-BiLSTM | BERT-BiLSTM-Attention | Improved Model |
|---|---|---|---|---|---|---|---|
| P/% | 71.17 | 76.35 | 78.31 | 89.61 | 90.17 | 94.24 | 96.71 |
| R/% | 73.49 | 75.86 | 74.84 | 86.12 | 88.49 | 89.61 | 95.88 |
| F/% | 72.31 | 76.1 | 76.54 | 87.83 | 89.32 | 91.87 | 96.29 |

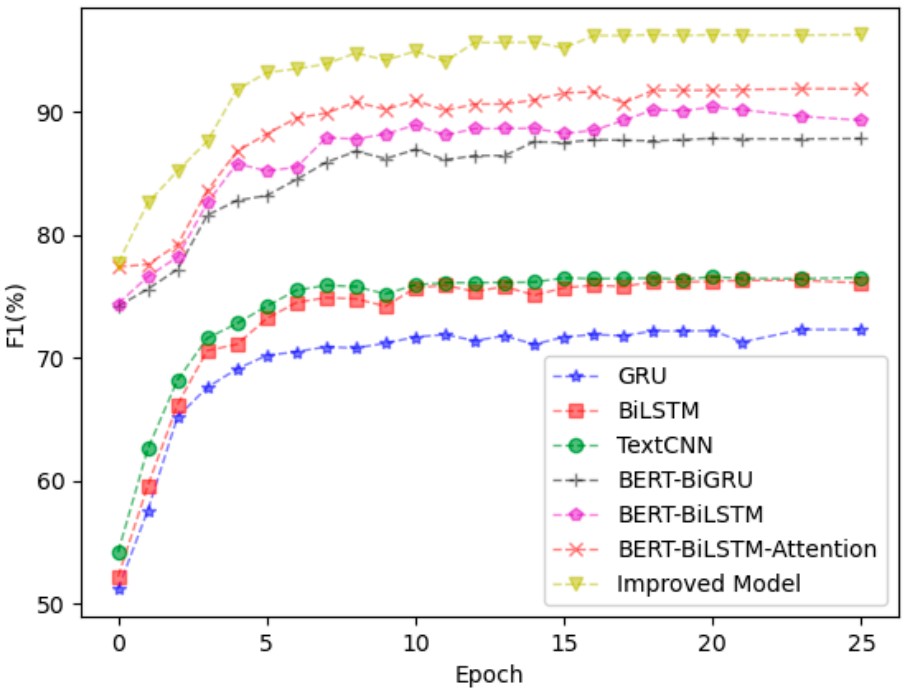

**Figure 12.** F1 value update.

4.4.3. Experimental Results of Multimodal Text Recognition and Sentiment Analysis Model

The experimental effects of this chapter are exemplified by the movie review video. Figure 13a,b shows the multimodal text recognition effect of the negative movie review video before and after improvement. Figure 14a,b shows the multimodal text recognition effect of the movie positive review video before and after improvement. Figure 15a,b shows the multimodal text recognition effect of the movie neutral review video before and after improvement. The results in Figures 13–15 show that three kinds of opinion sentiments in videos, namely, positive, neutral, and neg-ative, can be detected. In addition, the improved method in this paper can correctly identify more text in the video with higher recognition ability.

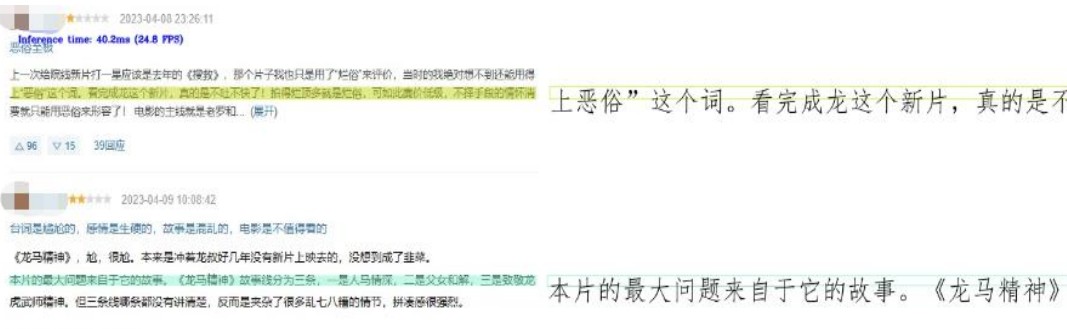

(**a**)

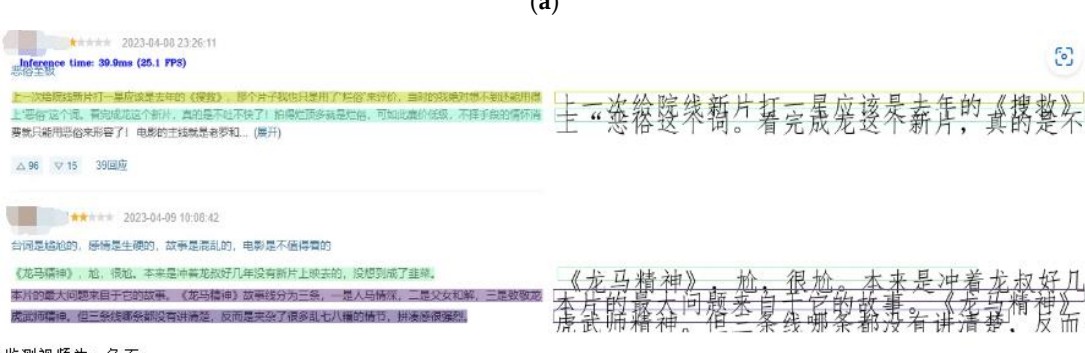

**Figure 13.** *Cont.*

**Translation of the content on the left side of the above image (b):**

Vulgar to the extreme

The last time I gave a star to a new theatrical film should have been last year's "Search and Rescue", the film I also just used "bad" to evaluate. At the time, I would never have thought that the word "vulgar" could be used. Look at the completion of the new film. It really is not vomit! Shooting bad at best is bad vulgarity, but so cheap and cheap, unscrupulous consumption of emotion can only be described as vulgar! The main line of the film is the old Luo and.

The lines are awkward, the feelings are raw, the story is confusing, and the movie is not worth watching.

The Spirit of Dragon Horse" is awkward, very awkward. It was originally intended for Uncle Long to go several years without a new film in theaters, but I did not expect to become a leek. The biggest problem with this film comes from its story. The story line of "The Spirit of Dragon and Horse" is divided into three lines: one is the love of people and horses; the second is the reconciliation of father and daughter; and the third is a tribute to the spirit of Dragon and Tiger martial arts masters. But which of the three lines is not clearly told but instead is interspersed with a lot of messy plot and a strong sense of patchwork.

**Translation of the content on the right side of the above image (b):**

The last time I gave a star to a new movie in theaters should be last year's "Search and Rescue.

On the word "vulgar". Look at the completion of the new film Dragon, really is not.

The spirit of the dragon horse", awkward, very awkward. Originally, it was for Uncle Long several.

The biggest problem with this film comes from its story. The Spirit of the Dragon and Horse.

The spirit of the tiger martial artist. But which of the three lines is not clearly stated, instead

(**b**)

**Figure 13. (a)** Improve the effect of multimodal text detection in a former negative movie review video. (**b**) Improved multimodal text detection effect for negative movie review video.

监测视频为：正面

**Translation of the content on the left side of the above image (a):**

The spirit of the dragon horse" in the parallel world of Jackie Chan himself ~

I do not know how many people and I grew up watching Jackie Chan's action movies ~ as a child super love his "Plan A" and "Drunken Fist" ~ his movies are on fire to the extent that they can be revisited every year on various channels. Over the years, he has made great contributions to the development of the Chinese film industry, but also a lot of well-known domestic and international film works. Until now, we can often see his films released, enough to prove that Jackie Chan's brother is an evergreen tree-like existence. Thanks to my childhood idol, he has been accompanied by his fans ~

**Figure 14.** *Cont.*

I'm glad to have another Jackie Chan movie to meet with you this year ~ "The Spirit of Dragon Horse." I went to see it for you first! In a word, we exceeded expectations! This movie has a special meaning to Jackie Chan; it is to commemorate his 60th anniversary in film and is a movie made from the heart. Director Yang Zi had revealed that the script was in its early stages and that Jackie Chan was his only choice for the film. Jackie Chan plays a down-and-out Dragon Tiger martial artist, Lao Luo, and his beloved horse, Red Rabbit, who relies on him to shoot thrilling action scenes on set for a living. Jackie Chan and the red hare "one man, one horse" fresh combination of not only laughs, tears are really poking me. "One man and a horse to keep the scene is particularly great, solid emotion, the arc is complete, the last is also very tearful, moved to the scene of every audience~

---

**Translation of the content on the right side of the above image (a):**

Very Changxing this year there is another Jackie Chan movie to meet with you!

The movie has a special meaning to Jackie Chan, it is to commemorate his career in film.

Duan, Jackie Chan is his only choice for this film. Jackie Chan in the film.

(**a**)

**Translation of the content on the left side of the above image (b):**

The spirit of the dragon horse" in the parallel world of Jackie Chan himself ~

I do not know how many people and I grew up watching Jackie Chan's action movies ~ as a child super love his "Plan A" and "Drunken Fist" ~ his movies are on fire to the extent that they can be revisited every year on various channels. Over the years, he has made great contributions to the development of the Chinese film industry, but also a lot of well-known domestic and international film works. Until now, we can often see his films released, enough to prove that Jackie Chan's brother is an evergreen tree-like existence. Thanks to my childhood idol, he has been accompanied by his fans ~

I'm glad to have another Jackie Chan movie to meet with you this year ~ "The Spirit of Dragon Horse." I went to see it for you first! In a word, we exceeded expectations! This movie has a special meaning to Jackie Chan; it is to commemorate his 60th anniversary in film and is a movie made from the heart. Director Yang Zi had revealed that the script was in its early stages and that Jackie Chan was his only choice for the film. Jackie Chan plays a down-and-out Dragon Tiger martial artist, Lao Luo, and his beloved horse, Red Rabbit, who relies on him to shoot thrilling action scenes on set for a living. Jackie Chan and the red hare "one man, one horse" fresh combination of not only laughs, tears are really poking me. "One man and a horse to keep the scene is particularly great, solid emotion, the arc is complete, the last is also very tearful, moved to the scene of every audience~

**Figure 14.** *Cont.*

Translation of the content on the right side of the above image (b):

I do not know how many people and I grew up watching Jackie Chan's action.
The film works that enjoy domestic and international fame can often be seen even now.
The movie has a special meaning to Jackie Chan, it is to commemorate his career in film.
Chibi Ma makes his living by shooting thrilling action scenes on set. Jackie Chan and.
A man and a horse to keep the scene interpretation is particularly great, solid emotion.

(**b**)

**Figure 14.** (**a**) Improve the effect of multimodal text detection in movie positive review video. (**b**) Improved multimodal text detection effect for movie positive review video.

Inference Time: 35.4ms/ (28.3 FPS)

中外影视作品中的亲子关系表达？

关于昨天看《速度与激情10》期间引申想到中外影视作品中亲子关系呈现方式的差异。

最近半年看下来的几个大片都有类似的感觉，相比于早年好莱坞作品对于人类命运、宇宙伦理等宏大叙事的渲染，疫情后《阿凡达2》《银河护卫队3》这几部电影，无一例外都在重视表达亲子/家庭组带下主角团的抉择与取舍。

不由想到，大多数中国影视作品中，一路的剧情转合与人物悲喜都只为奔赴一场Ending的亲子相聚或家族团圆；而西方更重视表达我们因血缘而产生伴，因陪伴而成长成为彼此的精神力量，我们在反派来临之时消释青春期因不理解而产生的前嫌，在相互守护成功后，又踏上各自的旅程。

最近半年看下来的几个大片都有类似的感觉，

圆；而西方更重视表达我们因血缘而产生伴，

监测视频为：中立

Translation of the content on the left side of the above image (a):

Representation of parent-child relationships in Chinese and foreign film and television productions.

About the differences in the way parent-child relationships are presented in Chinese and foreign film and television works that came to mind during yesterday's viewing of "Fast X".

The last six months to see down several blockbusters have a similar feeling, compared to the early years of Hollywood works for the fate of mankind, the universe ethics, and other grand narrative renderings. After the epidemic "Avatar 2" and "Guardians of the Galaxy Vol. 3" these films, without exception, are paying attention to the expression of parent-child or family ties under the protagonist group of choices and trade-offs.

I can't help but think that in most Chinese films and TV productions, the plot turns and the characters' sadness and happiness all the way to an ending parent-child reunion or family reunion. In the West, we attach more importance to expressing our bonding by blood, which grows to become each other's spiritual strength, and we release our youthful suspicions arising from lack of understanding when the villain comes, and then embark on our respective journeys after successfully guarding each other.

Translation of the content on the right side of the above image (a):

The last six months to watch down a few blockbusters have a similar feeling, the phase.
round; while the West places more importance on expressing that we are bound by blood, by

(**a**)

**Figure 15.** *Cont.*

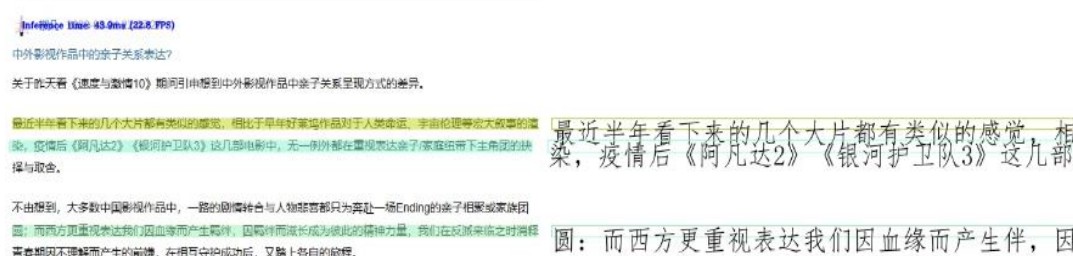

Translation of the content on the left side of the above image (b):

Representation of parent-child relationships in Chinese and foreign film and television productions.

About the differences in the way parent-child relationships are presented in Chinese and foreign film and television works that came to mind during yesterday's viewing of "Fast X".

The last six months to see down several blockbusters have a similar feeling, compared to the early years of Hollywood works for the fate of mankind, the universe ethics, and other grand narrative renderings. After the epidemic "Avatar 2" and "Guardians of the Galaxy Vol. 3" these films, without exception, are paying attention to the expression of parent-child or family ties under the protagonist group of choices and trade-offs.

I can't help but think that in most Chinese films and TV productions, the plot turns and the characters' sadness and happiness all the way to an ending parent-child reunion or family reunion. In the West, we attach more importance to expressing our bonding by blood, which grows to become each other's spiritual strength, and we release our youthful suspicions arising from lack of understanding when the villain comes, and then embark on our respective journeys after successfully guarding each other.

Translation of the content on the right side of the above image (b):

The last six months to watch down a few blockbusters have a similar feeling, the phase.

Infection, after the epidemic "Avatar 2" "Guardians of the Galaxy Vol. 3" these.

round; while the West places more importance on expressing that we are bound by blood, by

(**b**)

**Figure 15.** (**a**) Improving the effect of multimodal text detection in movie neutral review video. (**b**) Improved movie neutral review video multimodal text detection effect.

### 5. Conclusions

In this paper, the experiment we conducted will be used for dynamic monitoring of a large number of risk images and videos posted on relevant websites. This is mainly used to monitor websites with possible risky content in Henan Province, China, to help relevant departments monitor and manage risk websites.

We propose an improved visual multimodal text recognition and sentiment analysis method for scenarios such as online opinion analysis. In addition, the effectiveness of the method is experimentally verified. This paper has three main contributions: First, to improve the model for multiscale text detection and recognition, the LK-PAN network with a large sensory field is proposed to upgrade the CML distillation strategy, and an RSE-FPN with a residual attention mechanism is employed to enhance the feature map characterization ability. Second, the original CTC decoder is changed to the GTC method to improve the inference accuracy of the model for text recognition. A sinusoidal loss function for rotation recognition is also used, which can enhance the text detection performance for scenes with arbitrary rotation angles. Finally, we introduce the existing sentiment analysis model, integrate the BERT model, and improve the block attention mechanism. The original block attention mechanism method only took into account information about

global features. This paper adds information about local features and solves the problem that the original words with multiple meanings and word vectors easily lose the word position in the splicing process, which causes intermediate vectors to lose information in the encoding process. The improved model strengthens text recognition and sentiment distinctions in images and videos and may help in the management of internet public opinion departments. Meanwhile, we are currently integrating it into the public opinion platform of the laboratory.

Currently, the paper's sentiment analysis model is only applicable to the Chinese language. In the future, as English is an important language of communication today, we are going to add sentiment analysis models for English text. Adding English sentiment analysis to the model can improve the application scenarios of online public opinion initiatives. We will also add additional languages as required by the project at hand. Moreover, we will continue to optimize text recognition algorithms and pretraining models for sentiment analysis to provide crucial technical support for internet public opinion analysis in multimodal fields.

**Author Contributions:** Conceptualization, X.L., Q.Z. and F.W.; methodology, X.L., W.J. and F.W.; software, X.L., W.J. and F.W.; validation, Z.C.; formal analysis, Y.Q.; investigation, J.L.; resources, L.N.; data curation, X.L., J.L.; writing—original draft preparation, X.L.; writing—review and editing, H.D.; visualization, H.D.; supervision, F.W.; project administration, F.W.; funding acquisition, Q.Z. and F.W. All authors have read and agreed to the published version of the manuscript.

**Funding:** This work was supported by the National Natural Science Foundation of China (No. 62272163), Key Research Projects of Henan Higher Education Institutions (No. 23A520031), Open Foundation of Henan Provincial Key Laboratory of Network Public Opinion Monitoring and Intelligent Analysis (No. HNPOL202101002), Henan Province Science Foundation for Youths (No. 222300420230), and Open Foundation of Henan Key Laboratory of Cyberspace Situation Awareness (No. HNTS2022005).

**Institutional Review Board Statement:** Not applicable.

**Informed Consent Statement:** Not applicable.

**Data Availability Statement:** Not applicable.

**Conflicts of Interest:** The authors declare no conflict of interest.

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
