# Peer review of "MTR-SAM: Visual Multimodal Text Recognition and Sentiment Analysis in Public Opinion Analysis on the Internet"

_applsci, doi:10.3390/app13127307_

Round 1

Reviewer 1 Report

It will be helpful to outline the context of the presented analysis at the beginning - in particular, how much it is based on Chinese observations (social media, etc.), Asian and/or international in general. 

Moreover, what is more relevant is the language - it should be clearly stated at the beginning whether the methods for text recognition and subsequent sentiment analysis are based on English, Mandarin, Cantonese, or are these methods language independent. Literature review presents mostly methods applicable for English, but later on examples are mixed and this is unclear. Examples and tests are on mixed data - ICDAR 2015 dataset, then examples 13-16 are with Chinese data.

It needs to be discussed what is the definition of "risky video or image sentiment", what is positive, negative, neutral. Further, in 4.2 the categories (beside neutral and whatever is considered positive) are terror-related, violence-related, gambling-related, porn-related. It needs to be discussed and justified why this system of categories is chosen. Further, only negative sentiment categories are considered marked. This may be relevant to the reported improvement.

The particular features of the web crawler might also be relevant to the collected data and thus, to the training phase and results. Furthermore,  it is not clear how the title of the paper and namely, "Public Opinion Analysis on the Internet" is relevant to the content of the paper except in the improvement in speed (which is relevant if sentiment analysis is performed in real time) and in the specific process of web crawling. It would be beneficial to clearly discuss the specifics of the models and methods as applicable to analysing online content, and the specifics of the web crawling techniques.

At the beginning of section 3 there are multiple statements which show possible improvements: This network can accelerate the training speed (line 313), it can show good results for detecting text (line 319), etc. It is not clear which of these have been implemented and which were only discussed or suggested as improvements. 

More clarity is needed about the baseline methods in comparison to which the improvement of results is reported. Text recognition speed and accuracy is compared to PP-OCR as baseline (again, not clear which language models were used). Improvement of sentiment analysis needs more discussion. Improvement of 4.42% on sentiment analysis is not clearly significant.

Translation of text on Figures 13-16 would be helpful to see improvement. Beside that, it is unclear how the improved method in 4.4.3 works. 

It is also not clear how the problem with semantic disambiguation is solved (as stated in the conclusion) using the attention layer or otherwise.  

In summary, I think the paper would benefit from: (1) Clear presentation of types of modalities of multimodal content and what components exactly the authors aim to extract and how these contribute to the analysis; (2) more detailed description of the web crawling method and how the methodology is relevant for analysing online content (in real time, or not); (3) clear discussion on the selection of sentiment categories; (4) clearly stating the language(s) the model is applicable to and the results are based on.

Minor notes:

lines 349, 357, 458, 472, 473 and elsewhere
Some symbols in maths expressions are not properly rendered

lines 17-18 - repetitive
information in images and videos and discriminate sentiment, making quickly and accurately acquiring and identifying textual information in images and videos

lines 93-94 - repetitive
solve the problem of text recognition in complex scenes such as videos and effectively solve the shortcomings of the original model for poor detection of distorted images or videos

line 333 - repetitive
layer with residual structure by introducing a residual structure

line 483 - repetitive
crawlers are used in multimodal text recognition experiments to validate the model for text recognition

line 485 - repetitive
1000 randomly selected images in the ICDAR 2015 dataset, and 500 test images are taken randomly

line 536, 537 and elsewhere - for precision use F1
the F value of the text detection >  the F1 value of the text detection

If possible, use subscript F_1

line 548 - repetitive
the recognition ability of the improved model has been improved

line 550 - for precision and consistency
emotion analysis model > sentiment analysis model

Reviewer 2 Report

I believe this paper contributes to the literature. However, there are some problems in terms of presentation.

1. The authors should receive English proofreading and improve the readability.

2. Many issues were mentioned at the paper (e.g. lines 151-152, 166-168, 221-222). The authors should focus the main issue of the paper and reduce redundancy.

Due to these points, the contributions of the paper still vague. For example, the authors said that this paper contributes to three areas in line 111, but there was not explanations about "which area."

This paper must get proofreading.

Reviewer 3 Report

This study develop a new method of visual multimodal text recognition and sentiment analysis  20 (MTR-SAM) for web opinion analysis scenarios based on the most recent advances in the field of   text recognition. Autors dpropose a multimodal text recognition and sentiment analysis model MTR-SAM,  which not only improves the latest PP-OCR model but also introduces a direction-aware  function that optimizes the recognition of rotated fonts, enhances the detection of distor-tion and directional skew, and proposes a sentiment analysis model that enhances the  contextual relationship between words and improves the accuracy of discriminating the  sentiment polarity of text.

This approach seems to be extremely relevant and promising. I am very impressed with the results that the authors of the article have achieved.

Comments and Suggestions for Authors:

 ·               The authors indicate: « The training samples are taken from  1000 randomly selected images in the ICDAR 2015 dataset, and 500 test images are taken randomly. The video collected by the crawler is first cropped at the frame level using the  video editing tool Adobe Premiere Pro, the text with opinion information in the video is mainly selected, and each video does not exceed 30 seconds. Finally, 101 original videos  were collected, 3,261 video clips were acquired, and all video clips were annotated and  used as test data». Sample numbers do not seem too convincing.

·               The process of discussing the results can be extended by applying the results and extrapolating them to other similar studies.

Round 2

Reviewer 2 Report

The draft is improved. I agree with revision of authors.

I recommend authors to add discussion about generality of your work. Is your work Chinese dependency? Or this system can apply other languages?
